# Knowledge, attitudes, and practices associated with vitamin D supplementation: A cross-sectional online community survey of adults in the UK

Nuttan Kantilal Tanna[1,2�]*, Manisha Karki[3‡], Iman Webber[3‡], Aos Alaa[3‡], Austen El-Costa[3�], Mitch Blair[1,2�]

1 Department of Primary Care & Public Health, Imperial College London, London, United Kingdom, 2 River Island Paediatric and Child Health Academic Unit, Northwick Park Hospital, London North-West University Healthcare NHS Trust, London, United Kingdom, 3 Self-Care Academic Research Unit (SCARU), School of Public Health, Imperial College London, London, United Kingdom

উ These authors contributed equally to this work.
‡ MK, IW and AA also contributed equally to this work.
* nuttantanna@nhs.net

**Data Availability Statement:** All relevant data are within the manuscript and its Supporting information files.

## Abstract

### Objective

Assess knowledge, attitudes, and practices (KAPs) of a diverse population. Identify barriers and facilitators that inform routine vitamin D supplementation and self-care in the community setting.

### Design

Cross-sectional online voluntary survey. Electronic survey link published on college Qualtrics platform and advertised widely. Study information provided with Participant Information Sheet.

### Setting and participants

556 community dwelling adults across the UK.

### Methods

The overarching study included two phases, incorporating quantitative and qualitative methodologies. This paper reports findings from the first phase of the FABCOM-D (Facilitators and Barriers to Community (Healthy) Vitamin D status) study. Online survey questions were iteratively developed after background literature searches and piloted to ensure clarity and ease of understanding. Survey responses summarised using frequencies and percentages, and univariable and multivariable logistic regression models explored for any association. A p-value less than 0.05 was considered statistically significant. The Checklist for Reporting Results of Internet E-Surveys guided reporting. Statistical analysis performed using IBM SPSS software.

**Funding:** The author(s) received no specific funding for this work.

**Competing interests:** The authors have declared that no competing interests exist.

## Main outcome measures

Awareness of vitamin D information sources, health benefits and testing. Attitudes to supplementation, sun exposure and fortification.

## Results

Three quarters of the community had some awareness of vitamin D and around half were taking supplements. The two most trusted sources of information included health professionals and the NHS website. Participants were willing to pay for supplements, supporting a self-care agenda. With increasing age, there was significant reduced intake of vitamin D supplements. This aspect needs to be explored further as this could be a concern in deficiency status in the elderly. There was acceptance of food fortification but uncertainty on how to balance food intake with supplementation.

## Conclusion

We were successful in eliciting views on KAPs around vitamin D from a community population including a large proportion of Black and Minority Ethnic individuals. The community wanted information and guidance to help manage individual vitamin D status, especially for high-risk groups, and on balancing supplementation, food fortification and sun exposure.

## Introduction

Vitamin D, a fat-soluble steroid hormone precursor, is responsible for regulating calcium and phosphorous metabolism in humans [1] and essential for musculoskeletal health. Vitamin D deficiency (VDD), defined as a serum level of 25-hydroxy-vitamin D (25(OH)D) of less than 25 nmol/l, is a global health problem with over 1 billion people affected worldwide [2]. Over the last decade, there has been considerable interest in vitamin D and the use of vitamin D supplements to help prevent or treat a variety of medical presentations [3]. Claims in the media abound, and sales of vitamin D supplements have increased 10-fold since 2001 [3]. The risk of VDD is higher in pregnant women, children under five years of age, and in people without much sun exposure including frail and institutionalised individuals and ethnic groups with darker skin pigmentation (e.g. Asian and African populations) [4–6]. Adequate vitamin D status during pregnancy is important for foetal musculoskeletal development and general foetal growth [7], with a Cochrane review noting that the consequences of deficient levels during pregnancy could include lower neonate birth weight, head circumference and length [8]. Adequate vitamin D levels have been associated with lower risks of pre-eclampsia and gestational diabetes [8]. VDD also causes nutritional rickets; a bone disease seen in young children which can also be due to poor dietary calcium intake [9] and still prevalent in the UK despite public health campaigns. Chronically deficient levels in preschool children have also been associated with hypo-calcaemic seizures, and cardiomyopathy, as well as motor delay, aches, pains, and fractures [5, 10]. Vitamin D may also affect immune and cardiovascular system modulation [6]; however, studies have not yet demonstrated a clear causative effect [11–13]. Long-term health outcomes of VDD may include development of obesity, diabetes, asthma, hypertension, depression, osteoporosis with osteomalacia as an underlying condition increasing the risk of fragile bones [14, 15], neurodegenerative diseases and some cancers [8].

The main source of vitamin D in humans is solar radiation, scattered and filtered through Earth's atmosphere and obvious as daylight when the sun is above the horizon. Upon exposure to sunlight containing sufficient ultraviolet B (UVB) radiation, vitamin D is synthesised in the skin [11, 16]. Vitamin D can also be obtained from foods or dietary supplements. Dietary sources are essential when sunlight containing UVB radiation is limited, as is the case during the winter months in northern latitudes or where skin exposure is restricted [17]. Vitamin D initially goes through hepatic metabolization to form 25(OH)D. It is subsequently metabolised in the kidneys to calcitriol, a physiological active form of vitamin D. As calcitriol has a very short half-life in the plasma, levels of its precursor, 25(OH)D, are measured to assess vitamin D status [18, 19]. In the UK, a daily supplement of 10 micrograms per day (400 international units per day) is recommended for those aged four years and above [20]. People with higher body mass index (BMI) and larger waist circumferences have lower serum levels [21], and higher doses of vitamin D supplements may be needed.

Previous research, including questionnaire studies [14, 22, 23], has been conducted to assess the knowledge, attitudes, and practices (KAPs) in relation to Vitamin D. In a Scottish study conducted to assess the KAPs of vitamin D, around 90% of the study population was Caucasian, and many other groups traditionally considered to be at risk of vitamin D deficiency were not well represented [23]. There is a pressing need to study the knowledge, attitudes, and practices of the wider community, in particular from Black, Asian and minority ethnic sub-populations, to gain an understanding of the facilitators of and barriers to vitamin D supplementation in men, women, and children. Lee C et al explored these aspects in depth with a Somali community in Northwest London and found that a lack of awareness, access, and a reluctance to medicalise a natural state (pregnancy) were key issues [10, 24]. The work undertaken included a systematic review that reported a paucity of studies that address the behavioural determinants that would help attain an adequate vitamin D status in hard-to-reach communities [10].

This paper reports findings from the online survey, the first phase of the FABCOM-D (Facilitators and Barriers to Community (Healthy) Vitamin D status) study. It was undertaken to assess the KAPs of the diverse community in the UK, including both Caucasian and minority ethnic groups and offers important insights on the current usage of vitamin D and associated issues.

## Study objectives

The objective of the FABCOM-D study is to investigate the barriers and drivers for the routine supplementation of vitamin D in the community setting. We first explore where and how people receive their health information on vitamin D and the benefits of supplementation. The study design includes two phases, a cross-sectional online survey with findings related to knowledge, attitudes, and practices reported in this paper. Further in-depth study of the topic within focus group and interview settings in the second phase will add to our knowledge on what the facilitators and barriers are.

## Methods

### Study design

A cross-sectional online survey, incorporating iteratively designed questions, was implemented to assess attitudes and experiences regarding Vitamin D and supplementation within the respondent community. The electronic survey (e-survey) link was published and available on the Imperial College Qualtrics platform between 10 February and 02 September 2021 (7 months). The survey was disseminated by email using convenience sampling

through existing personal and professional networks, including ARC (Applied Research Collaboration) Northwest London, the CHAIN Network (Contact, Help, Advice, and Information Network) and social media platforms. It was open to all and could be accessed by anyone with a link to the web portal. Study information was provided and included a Participant Information Sheet (PIS). The PIS included information on study aims, protection of participants' personal data, including their right to withdraw from the study at any time, which data were stored, where and for how long, who the investigators were, and survey length. Participants were informed that this was a voluntary survey without any monetary incentives but that the aim was to publish study results. Participants were also asked if they would be interested in being part of a focus group or one to one interview in the study's second phase, with the opportunity to discuss these issues more fully. The potential collective benefits of taking part in terms of helping advance knowledge in this area was highlighted. The data collected was stored on the Imperial College London secure database, and only the research team could access the e-survey results.

## Electronic survey

The e-survey contained 36 questions and automatically captured responses. The questions were agreed with consensus by the research group via iterative development and review process rounds. The research group also included medical students who had opted to work with the RI-PAC unit's life course Vitamin D project team, for experiential formative learning, on their research speciality choice module at Imperial College, London [25]. Before answering the e-survey questionnaire, participants were asked to confirm their consent, via the electronic survey link. The questions were displayed on pages accessible using either a personal computer or smartphone. Questions on respondent demographic characteristics included gender, age, ethnicity, and occupation/field of work, the first part of postal code, dietary restrictions, skin type [26] and medical conditions. Participants could review their answers again before submitting them. The survey did not contain any conditional questions. Survey responses were only excluded if much of the survey was incomplete. All data collected through the survey were anonymised and not personally identifiable. The online survey was piloted with a small group of individuals to ensure technical functionality and usability before being published. The e-survey questions were informed by previous background literature searches and included asking about the participant's own experience with Vitamin D, potential health benefits, and who the respondents' thought were in at-risk categories. The survey also covered preferences for supplementation, food fortification, testing and attitudes to pricing (see S1 File for full copy of survey questionnaire). Data were collected with the e-survey questionnaire administered on Qualtrics, using web-based software. Qualtrics's websites have first party cookies and allow third parties to place cookies on devices. As no IP addresses were collected, the team could not identify any cases of duplicate entries.

## Statistical analysis

Survey responses were summarised using frequencies and percentages. Univariable and multivariable logistic regression models explored the association. A p-value less than 0.05 ($<0.05$) was considered statistically significant. The Checklist for Reporting Results of Internet E-Surveys (CHERRIES) was used to guide reporting [27]. The statistical analysis was performed using IBM Corp. Released 2020. IBM SPSS Statistics for Windows, Version 27.0. Armonk, NY: IBM Corp.

### Ethics statement

The study was given ethical approval by Imperial College Research Ethics Committee (ICREC # 20IC6433) on 08/02/2021.

## Results

### Demographic profile of respondents

The e-survey captured responses from some 557 respondents from across the United Kingdom: with 71 excluded due to missing data for age, sex, and ethnicity. Table 1 presents the demographic data with frequency and percentage figures for the total of 486 respondents included for analysis.

The majority of respondents were women (356; 73%), with just over a quarter male respondents. Most (136;28%) were aged between 51–60, with 115 (24%) indicating that they were between 61–70 and 89 (18%) between 31–40 years of age. Nearly half (240;49%) of respondents were from a White ethnic background, with 221 (45%) combined from an Asian (29%) and British Black/African/Caribbean (16%) background (see Table 1). Over three-quarters (372;76%) of respondents said they had a university degree or higher, and more than half (257;53%) had a full-time job. 78 (16%) had retired.

116 (33%) female respondents were in the menopausal phase. Over half of the respondents (275;57%) claimed they did not have any dietary requirements. 80 respondents (17%) stated that they were vegetarian whilst 50 (10%) followed a halal diet. Other recorded diets were included for analysis; however, few respondents chose them. The e-survey included Fitzpatrick-informed [26] pictorial and descriptive skin colours. 141 respondents, that is nearly a third (29%) of the respondents coded themselves as being Type 3 (Medium, white to olive), with 24% (117) saying they had Type 2 (White, fair) skin type and colour whilst 20% (96) coded themselves as Type 4 darker skin colouring (Olive, moderate brown).

There was no significant association between Vitamin D intake and ethnicity (p-value 0.68), BMI (p-value 0.80; not all respondents provided their BMI data), medical condition (p-value 0.20) (medical condition regrouped (p-value 0.91), and skin type (p-value = 0.25); however, statistically significant association was seen with vitamin D intake and gender (p-value 0.05), age (p-value <0.01), sun exposure(Q23) (p-value 0.05), and skin coverage (Q24) (p-value <0.01) (Table 1).

### Awareness of vitamin D information sources, benefits, and testing

To understand the level of the community's knowledge on vitamin D, seven multiple-choice questions were included in the e-survey. When asked where they had heard of vitamin D in the past, around three quarters (75%) indicated that they had known about it generally, whilst just over 60% said they had received their information from health professionals, with under half equally saying that they had heard about vitamin D from family and friends (45%) or the media (43%).

When asked what the two most useful sources of health information were, over 70% indicated these to be health professionals (371;77%) to include doctors, nurses and pharmacists, and the NHS website (343;71%) as their preferred options. Around a third (32%) used the internet or other media to include TV, newspapers, radio, internet, and magazines, for health information.

Respondents perceived vitamin D to have the following health benefits: around 80% felt that vitamin D helped support the immune system (399;82%) and prevented osteoporosis or

**Table 1. Association between Vit-D intake (Q.9—see S1 File) and age, gender, ethnicity, BMI, skin type, sun exposure and skin coverage.**

| | Vit-D supplement intake | | | p-value |
|---|---|---|---|---|
| | Yes n (%) | No n (%) | Total N (%) | |
| **Gender** | | | | **0.05** |
| Male | 96 (24.8) | 34 (34.3) | 130 (26.7) | |
| Female | 291 (75.2) | 65 (65.7) | 356 (73.3) | |
| **Age** | | | | **<0.01** |
| 20–30 | 35 (9.0) | 16 (16.2) | 51 (10.5) | |
| 31–40 | 61 (15.8) | 28 (28.3) | 89 (18.3) | |
| 41–50 | 59 (15.2) | 15 (15.2) | 74 (15.2) | |
| 51–60 | 109 (28.2) | 27 (27.3) | 136 (28.0) | |
| 61–70 | 106 (27.4) | 9 (9.1) | 115 (23.7) | |
| 71 and over | 17 (4.4) | 4 (4.0) | 21 (4.3) | |
| **Ethnicity** | | | | **0.68** |
| White | 186 (48.1) | 54 (54.5) | 240 (49.4) | |
| Mixed/Multiple ethnic groups | 20 (5.2) | 5 (5.1) | 25 (5.1) | |
| Asian/Asian British | 118 (30.5) | 25 (25.3) | 143 (29.4) | |
| Black/African/Caribbean/Black British | 63 (16.3) | 15 (15.2) | 78 (16.0) | |
| **Body mass index (BMI)** | | | | **0.80** |
| Underweight | 11 (3.6) | 4 (5.1) | 11 (3.6) | |
| Normal weight | 140 (45.3) | 38 (48.7) | 178 (46.0) | |
| Overweight | 109 (35.3) | 26 (33.3) | 135 (34.9) | |
| Obese | 49 (15.9) | 10 (12.8) | 59 (15.2) | |
| **Medical conditions** | | | | **0.20** |
| Liver problems | 9 (2.4) | 2 (2.0) | 11 (2.3) | |
| Kidney problems | 31 (8.2) | 12 (12.1) | 43 (9.0) | |
| Coeliac disease/Crohn's disease/ulcerative colitis | 12 (3.2) | 0 (0.0) | 12 (2.5) | |
| No, I don't have any of these | 327 (86.3) | 85 (85.9) | 412 (86.2) | |
| **Regrouped medical conditions** | | | | **0.91** |
| Absent | 327 (86.3) | 85 (85.9) | 412 (86.2) | |
| Present | 52 (13.7) | 14 (14.1) | 66 (13.8) | |
| **From the skin types (colours) in this picture, which skin type do you think that best describes your skin colour?** | | | | **0.25** |
| Type 1 LIGHT, pale white—Always burns, never tans | 27 (7.0) | 12 (12.1) | 39 (8.0) | |

(*Continued*)

**Table 1.** (Continued)

| | Vit-D supplement intake | | | p-value |
|---|---|---|---|---|
| | **Yes n (%)** | **No n (%)** | **Total N (%)** | |
| Type 2 WHITE, fair (Usually Burns, Tans with difficulty) | 90 (23.3) | 27 (27.3) | 117 (24.1) | |
| Type 3 MEDIUM WHITE TO OLIVE (Sometimes mild burns, gradually tans to Olive | 114 (29.5) | 27 (27.3) | 141 (29.0) | |
| Type 4 OLIVE, moderate brown (Rarely burns, Tans with ease to a Moderate Brown) | 83 (21.4) | 13 (13.1) | 96 (19.8) | |
| Type 5 BROWN, dark brown (Very rarely burns, Tans very easily) | 53 (13.7) | 16 (16.2) | 69 (14.2) | |
| Type 6 BLACK, very dark brown to black (Never burns, Tans very easily, deeply pigmented) | 20 (5.2) | 4 (4.0) | 24 (4.9) | |
| **Number of hours a day spent on the average outside in the sunlight in the spring & summer month** | | | | **0.05** |
| Less than 1 hour | 152 (39.3) | 33 (33.3) | 185 (38.1) | |
| Between 1–3 hours | 195 (50.4) | 46 (46.5) | 241 (49.6) | |
| Between 3–5 hours | 30 (7.8) | 16 (16.2) | 46 (9.5) | |
| More than 5 hours | 10 (2.6) | 4 (4.0) | 14 (2.9) | |
| **On average how much do you cover up during the spring/summer months?** | | | | **<0.01** |
| Minimal coverage (exposure of shoulders and above the knee) | 54 (14.0) | 29 (29.3) | 83 (17.1) | |
| Moderate coverage (exposure of forearms, below knee and face) | 274 (70.8) | 61 (61.6) | 335 (68.9) | |
| Maximum coverage (exposure only to hands and face) | 59 (15.2) | 9 (9.1) | 68 (14) | |

brittle bones (379;78%), with over 60% indicating that vitamin D could help prevent rickets (308;64%), also often colloquially referred to as "soft bones" in children. Some 480 (99%) respondents thought that people could increase their vitamin D levels with sunlight exposure, with 430 (89%) considering this could be achievable by taking supplements. Some 76% (369) felt dietary intake could help in this regard. When asked for their opinion on the two best ways to increase vitamin D levels, respondents chose sunlight and supplements (94% and 70% respectively).

Respondents were asked about who they thought would be at a higher risk of low levels of vitamin D, with most (87%) selecting option 'people who do not spend a lot of time outside during the day', with 76% (367) indicating 'people who cover up the majority of their skin when they are outside' and around 307(63%) choosing those with dark skin. When questioned about factors affecting vitamin D levels, most (90%) chose 'not getting enough sunlight', with 82% choosing the option 'not spending time outside during the day'.

When asked what the recommended daily intake of vitamin D in the UK is, a third of the respondents (34%) said this was 10 micrograms (400 international units), with a further 27% indicating 25 micrograms (1000 international units) but with nearly a quarter of respondents (24%) saying that they did not know.

Half of the respondents (51%) had previously had a blood test to check their vitamin D levels, with 40% saying that they had not had a test. Around 9% were not sure. When asked if they thought vitamin D testing should be part of the regular NHS health checks, the majority (425;91%) selected the option 'agree'. Respondents (388;80%) also agreed with the statement

that those who are in the risk category of vitamin D deficiency (such as older patients, pregnant women & people with dark skin tones) should get regular testing.

## Sun exposure

Half of the respondents (241;50%) spend on average 1–3 hours in the sunlight in the spring and summer months, whereas around 185 (38%) said this was less than 1 hour for them. The majority (70%) indicated usually having moderate coverage (exposure of forearms, below knee and face); just under a fifth (17%) said that they had minimal coverage (exposure of shoulders and above the knee). 66 (14%) respondents coded themselves as having maximum coverage, with exposure of only their hands and face, of whom 59 said that they took a vitamin D supplement (Table 1).

There was a statistically significant association between ethnicity and skin colour (p-value 0.01) in our respondent sample, but no significant association between hours of sun exposure and skin type (p-value 0.63), and Vitamin D supplement intake (p-value 0.20) (Table 2). Skin coverage was seen to be significantly associated with ethnicity (p-value <0.01), skin colour (p-value 0.04), and Vitamin D intake both for those who had taken supplements over the last year or for longer (p-value <0.01). With statistically significant association between vitamin D intake and gender (p-value 0.05), age (p-value 0.01), sun exposure (p-value 0.05) and skin coverage (p-value <0.01) (Table 3), logistic regression analysis was undertaken to assess the effect

**Table 2. Association between ethnicity, skin colour and Vit-D supplement intake with hours of sun exposure.**

| | Total N (%) | Hours of sun exposure (Q.23) | | | | |
|---|---|---|---|---|---|---|
| | | Less than 1 hour | Between 1–3 hours | Between 3–5 hours | More than 5 hours | p-value |
| | | n (%) | n (%) | n (%) | n (%) | |
| **Vit-D supplement intake** | | | | | | 0.20 |
| Yes recently (in the last 12 months) | 205 (42.2) | 76 (41.1) | 108 (44.8) | 16 (34.8) | 5 (35.7) | |
| Yes, for a number of years | 182 (37.4) | 76 (41.1) | 87 (36.1) | 14 (30.4) | 5 (35.7) | |
| No | 99 (20.4) | 33 (17.8) | 46 (19.1) | 16 (34.8) | 4 (28.6) | |
| **Vit-D supplement intake** (Regrouped) | | | | | | **0.05** |
| Yes | 387 (80) | 152 (82.2) | 195 (80.9) | 30 (65.2) | 10 (71.4) | |
| No | 99 (20) | 33 (17.8) | 46 (19.1) | 16 (34.8) | 4 (28.6) | |
| **Ethnicity** | | | | | | **<0.01** |
| White | 239 (45.2) | 69 (37.3) | 134 (55.6) | 29 (63.0) | 8 (57.1) | |
| Mixed/Multiple ethnic groups | 14 (2.6) | 10 (5.4) | 11 (4.6) | 3 (6.5) | 1 (7.1) | |
| Asian/Asian-British | 135 (25.5) | 70 (37.8) | 57 (23.7) | 12 (26.1) | 4 (26.1) | |
| Black/African/Caribbean | 78 (14.7) | 36 (19.5) | 39 (16.2) | 2 (4.3) | 1 (7.1) | |
| **Skin colour** | | | | | | 0.63 |
| Type 1 light, pale white | 39 (8) | 15 (8.1) | 19 (7.9) | 4 (8.7) | 1 (7.1) | |
| Type 2 white, fair | 117(24.1) | 42 (22.7) | 63 (26.1) | 10 (21.7) | 2 (14.3) | |
| Type 3 medium, white to olive | 141 (29) | 46 (24.9) | 72 (29.9) | 16 (34.8) | 7 (50.0) | |
| Type 4 olive, moderate brown | 96 (19.8) | 43 (23.2) | 40 (16.6) | 10 (21.7) | 3 (21.4) | |
| Type 5 brown, dark brown | 69 (14.2) | 30 (16.2) | 32 (13.3) | 6 (13.0) | 1 (7.1) | |
| Type 6 black, very dark brown to black | 24 (4.9) | 9 (4.9) | 15 (6.2) | 0 (0.0) | 0 (0.0) | |

• No significant association was found between the reported hours of sun exposure and skin colour (p-value 0.63), and Vitamin D supplement intake (p-value 0.20). This suggests that the hours of sun exposure reported by the respondents are not related to these variables.

• However significant association could be seen with ethnicity (p-value <0.01) and Vitamin D intake when the responses for taking Vitamin D question were regrouped together to include having taken Vitamin D supplements over the last one year, or for a longer period of time (p-value 0.05).

**Table 3. Association between ethnicity, skin colour and Vit-D supplement intake with skin coverage.**

| | Skin Coverage (Q.24) | | | | |
|---|---|---|---|---|---|
| | Total N (%) | Minimal coverage | Moderate coverage | Max coverage | p-value |
| | | n (%) | n (%) | n (%) | |
| **Vit-D supplement intake** | | | | | **<0.01** |
| Yes recently (in the last 12 months) | 205 (42.2) | 26 (31.3) | 153 (45.7) | 26 (38.2) | |
| Yes, for a number of years | 182 (37.4) | 28 (33.7) | 121 (36.1) | 33 (50.0) | |
| No | 99 (20.4) | 29 (34.9) | 61 (18.2) | 9 (13.2) | |
| **Vit-D supplement intake** (Regrouped) | | | | | **<0.01** |
| Yes | 387 (80) | 54 (65.1) | 274 (81.8) | 59 (86.8) | |
| No | 99 (20) | 29 (34.9) | 61 (18.2) | 9 (13.2) | |
| **Ethnicity** | | | | | **<0.01** |
| White | 239 (45.2) | 41 (49.4) | 179 (53.4) | 20 (29.4) | |
| Mixed/Multiple ethnic groups | 14 (2.6) | 9 (10.8) | 12 (3.6) | 4 (6.1) | |
| Asian/Asian-British | 135 (25.5) | 19 (22.9) | 94 (28.1) | 30 (45.5) | |
| Black/African/Caribbean | 78 (14.7) | 14 (16.9) | 50 (14.9) | 14 (21.2) | |
| **Skin colour** | | | | | **0.03** |
| Type 1 light, pale white | 39 (8) | 3 (3.6) | 28 (8.4) | 8 (11.8) | |
| Type 2 white, fair | 117 (24.1) | 14 (16.9) | 91 (27.2) | 12 (17.6) | |
| Type 3 medium, white to olive | 141 (29) | 33 (39.8) | 91 (27.2) | 17 (25.8) | |
| Type 4 olive, moderate brown | 96 (19.8) | 16 (19.3) | 66 (19.7) | 14 (21.2) | |
| Type 5 brown, dark brown | 69 (14.2) | 13 (15.7) | 40 (11.9) | 16 (24.2) | |
| Type 6 black, very dark brown to black | 24 (4.9) | 4 (4.8) | 19 (5.7) | 1 (1.5) | |

• Skin coverage was seen to be significantly associated with ethnicity (p-value <0.01), skin colour (p-value 0.03), and Vitamin D intake (p-value 0.01), and Vitamin D intake regrouped to include both those having taken Vitamin D supplements over the last one year or for a longer period of time (p-value <0.01).

of age on vitamin D intake, with adjustment for gender and ethnicity. This showed that with every one unit increase in age, with each unit equating to a 10-year period, there was likely to be a 30% decrease in vitamin D supplement intake (adj. OR = 0.70,95% CI (0.59–0.83).

## Supplements

Around 42% had taken vitamin D supplements within the last 12 months, just over a third (37%) had been taking them for some years, but with a fifth (20%) saying they did not take supplements. More than half (57.0%) of respondents were taking other vitamins or supplements; including calcium, collagen, hair, skin & nail, iron, magnesium, combined multi-vitamins, Omega 3, rosehip, vitamin B, B12, C, D & E and zinc. Over half (51.4%) of the respondents would be prepared to pay '*less than £5*' for vitamin D supplements for a month, with around a third (35.3%) indicating that they would pay between £5–10 for a month's supply. Respondents selected '*strongly agree*' and '*agree*' when reviewing the following statements: '*People at risk of vitamin D deficiency should get free vitamin D supplements*' and '*Doctors should check vitamin D levels before recommending supplements*' but did not think people should pay for their vitamin D supplements regardless.

When asked which factors were important to them when choosing vitamin D supplements, responses included aspects such as more information about the benefits of vitamin D, advice from health professionals (see Box 1), easy access to the doses required and the cost of supplements. Pharmaceutical formulation modifications such as the supplement

> **Box 1. E-survey respondents were asked which factors were important to them when choosing vitamin D supplements; the following were selected as '*Very Important or Important*':**
>
> - Knowledge about the health benefits of vitamin D
> - Advice from health professionals
> - Low vitamin D levels on blood tests
> - Experiencing symptoms of low vitamin D levels
> - Reduced exposure to sunlight
> - Easy access to the supplement
> - Access to the appropriate dosage over the counter (without prescription)
> - Cost of supplementation
> - How often I need to take the supplement.

containing other vitamins, nutrients & minerals, or the taste, flavour, and smell of the supplement or how easy the supplement was to chew, or swallow were classified as '*neutral or important*'. Whether the supplement was available in liquid form was classified as '*neutral and unimportant*'.

## Fortified foods

Many respondents (62%) said they would prefer to have foods that are fortified with vitamin D instead of taking separate supplements. When asked for what reason they might not want foods with added vitamin D, over 40% (42%) said they did not like the idea of eating processed or fortified foods, and around 16% thought it would be too expensive. A fifth of respondents (22%) provided further reasons using free text on why they would rather not have fortified foods (see Box 2). These included a preference to supplement, unable to or did not include standard fortified foods that were currently available within their diet, being unsure about the dosage of vitamin D within fortified foods and wanting the choice to make their own decisions.

## Discussion

This cross-sectional e-survey generated a wide range of helpful findings. Our respondent profile reflects the views of a female dominant, elderly, generally healthy and educated cohort. Around 50% classified themselves as overweight or obese, with 14% of respondents indicating that they had chronic ill health. We were successful in eliciting views on knowledge, attitudes, and practices around vitamin D from a diverse community population, which included a large proportion of Black and minority ethnic individuals, via this online survey questionnaire research study. There was a high level of awareness of vitamin D and its benefits within the community. We received feedback on issues ranging from valued sources of information, benefits, and testing for vitamin D levels, on sun exposure and sunlight as a source of vitamin D and perceptions of the at-risk populations, and with views provided on testing, supplementation, and fortification. Many of our respondents were already taking supplements or therapeutic doses related to their underlying illnesses. When, how, for what duration, and the dose that Vitamin D supplementation should be recommended for, for beneficial outcomes is a common, often posed question in clinical practice. The answer will vary depending on whether this is being considered in the context of population wide public health advice versus

### Box 2. Barriers towards fortified foods—free text quotes.

| | |
|---|---|
| **Preference to supplement** | • Because the amount of intake could be variable from day to day. It is easier to take a supplement and then top up with food<br>• Can regulate exactly how much I get from taking a supplement, would worry I wasn't getting enough from fortified foods<br>• Easier to eat normal food and take the supplement<br>• Easier to increase intake through supplementation if the type of foods that are fortified are not part of your normal daily diet<br>• Easier to just take a tablet & then you know how much you have taken<br>• I don't like some of the foods that would be fortified with vitamin D and would know I was getting the right amount through tablets<br>• I think supplements will also be required in addition to foods containing vit D<br>• Not as good as taking supplements<br>• It's far more convenient to take one supplement with the exact dosage you need. |
| **Unable or don't eat the standard fortified foods** | • Being diabetic, may not be able to have those certain items of food that has vit D, also being vegetarian, also I need very high doses of vit D and foods with vit D would probably not be enough (difficult to calculate each item every day)<br>• Dependant on what foods are fortified. I don't drink orange juice regularly as it contains a lot of sugar. I also don't eat a lots of bread and only eat wholemeal.<br>• I don't want to buy products that I wouldn't buy regularly (e.g., orange juice) just because of the addition of Vitamin D<br>• I have to careful what I eat, e.g orange juice is high in sugar, and I am a diabetic<br>• Unable to drink milk or orange juice<br>• It would limit my choosing option and may affect price<br>• I follow Keto eating plan and fortified foods may not be suitable<br>• I have absorption issues |
| **Unsure about dosage in fortified foods** | • Harder to ensure getting enough vitamin d<br>• Harder to manage a households vitamin needs<br>• I don't know if it has as much vit d as a supplement would<br>• I would like to know how much I am taking and be in control.<br>• I would not be able to track my vitamin d level intake a day if multiple foods have vitamin D as well<br>• I would be unsure if I would hit the required daily requirements.<br>• I would rather take my own dosage and then I know I have had the right amount<br>• I wouldn't know how to ensure I'm getting my recommended supplement dose<br>• Unlikely to be a high enough dose. You wouldn't know how much the foods cumulatively would add up to |
| **Wanting to make own decision** | • I think it should be down to an individual and their choice not forced upon them in products per say.<br>• I prefer to have a choice and make my own decisions about my health rather than be coerced<br>• Individuals should be able to choose<br>• I prefer to make my own decision<br>• I probably already eat fortified cereals, I just have no preference<br>• I want to know what I am taking |

---

### Box 3. Key messages.

---

Awareness about vitamin D was high within our diverse community respondent sample
- All respondents (99%) indicated that sun exposure was a source for vitamin D, and many felt that the two best ways to increase vitamin D levels were sun exposure (94%) and supplementation (70%)
- Trusted sources for vitamin D information rated highly included health professionals and the NHS Website
- A good proportion were willing to pay for vitamin D supplements
- Around 50% respondents were prepared to pay up to £5 for month's supply and a good third (35%) were prepared to pay between £5-£10 per month
- With increasing age there was reduced intake of vitamin D supplements, which could be a concern in deficiency status often seen in the elderly
- There was general acceptance of food fortification but confusion on how much vitamin D would be available from food sources and how to balance intake with supplemented doses

---

supporting '*patients*' as individuals with a medical condition where their vitamin D status needs to be addressed as part of their holistic management plan [4, 15, 28]. At a population level, this community wanted clear guidance on vitamin D status, and information to balance vitamin supplementation, food fortification and sun exposure. Box 3 presents key messages, based on findings from this community online survey.

### The way forward: Supplementation, fortification, or safe sun exposure?

Awareness of vitamin D was high in our community sample, so current public health messaging is getting some results; but there still was some confusion on vitamin D doses needed and how to balance this with sunshine exposure and dietary intake.

**Supplementation.** *Vitamin D supplementation*. Our 'healthy' e-survey respondents provided useful feedback by classifying various factors (see Box 1) as important when considering supplementation. These included improving knowledge about the health benefits of vitamin D, with advice provided by health professionals as key, assessing for low vitamin D levels by testing, and addressing symptoms of low vitamin D levels and reduced exposure to sunlight. These criteria should inform public health information leaflets. Pertinent to purchase of supplements were aspects such as easy access, availability at the appropriate dosage over the counter (without prescription) and cost. Respondents did not feel that pharmaceutically oriented formulation issues were of concern, and this maybe as these have already been addressed by manufacturers; for example, vitamin D supplements can be combined with other vitamins, nutrients and minerals, and taste, flavour, smell or how easy the supplement is to chew or swallow and availability in liquid form has been addressed, with a range of products available from pharmacies or health food stores.

*Affordability of supplements*. Interestingly on the issue of costs, with reference to purchasing vitamin D supplements, although half of our respondents said that they would be prepared to pay less than £5 for a month's supply, a good third (35%) was prepared to pay between £5-£10 per month. The wish to have some control over dosage, which is feasible when taking supplements, and to make one's own choice were key issues highlighted by the respondents (Box 2 presents a summary of open text statements from respondents).

The problem with regards to affordability has been compounded by recent changes to policy for prescriptions of vitamin D. Recent government advice to health professionals in England is that it is appropriate to request that patients increase or maintain their vitamin D levels in range by purchasing over the counter (OTC) supplements at their own expense [29], in most cases. Although this may help reduce the NHS financial burden, this shifts the burden

of vitamin D supplementation onto patients. In the context of large families, increasing unemployment, low incomes and cuts to social support, many individuals/families, may forego vitamin D supplementation in favour of other expenses they deem more essential. Consequently, rates of VDD may rise. This maybe a particularly severe issue for black and minority ethnic populations who are disproportionately on low income, not in employment and reliant on social support [30]. In work undertaken with the Somali community, Lee C et al also reported that many in the community felt that if their doctor did not provide a prescription, then in line with perception that they had a healthy diet, they would not purchase vitamin D supplements [24]. Our findings also suggest that as the community gets older, there is a reduction in use of vitamin D supplements; with around 30% reduction in vitamin D intake with every increase in age units of 10 years. This may be due to elderly people not being able to afford buying vitamin D supplements or alternatively they may have these being prescribed. We did not ask about prescribed supplements in people aged 60 and over, when there are no NHS prescription charges applied; but generally, we would have expected the respondent to still indicate that they were taking supplements, whether purchased or prescribed.

*Fortification.* Considering the issue of fortification, Rajwar E et al (2020) [31] recently conducted an overview of systematic reviews and reported a lack of robust evidence to support fortification of food with vitamin D and calcium for public health benefits; their review focused on the micronutrient's vitamin A, vitamin D and calcium. Their studies targeted women of reproductive age. However, many proponents have and do argue in favour of fortification [6, 32, 33] and in industrialised countries the routine fortification of certain staple foods with vitamin D is common practice [34, 35]. These foods include milk, cereal, juice, bread, yogurt, and cheese [36, 37]. Margarine fortification was mandatory in Denmark until 1985, when a political decision led to policy discontinuation [38]. Duus KS et al (2021) [38] noted that around 13% of the population's vitamin D intake at the time had been from fortified margarine. In the UK, margarine fortification was mandated between 1940 to 2013 [39]. Work undertaken to study populations living in India with its tropical climate [40] indicates that Indian diets, particularly where a vegetarian diet is followed, that these do not meet the daily requirement of vitamin D for a normal adult. The research group [40] recommended the establishment of national programs directed at policy makers in India, to consider fortification of various foods. Fortified foods available in India include *Vanaspati Dalda* ghee, manufactured from hydrogenated and hardened vegetable oils and fortified with 200 international units of vitamin D per 100 grams and certain branded milk products [40]. In western dwelling South Asian populations, food fortification of chapatti flour and increased use of vitamin D supplements has been recommended [41]. Food fortification could be a cost-effective way forward, as a preventative strategy [42], aiming to reduce the prevalence of vitamin D deficiency status in a population.

An important question is how much should staple food items be fortified with? A systematic review [37] with high evidence quality, showed that food fortification improved 25(OH)D concentration by a mean difference of 15.51 nanomole / litre, resulting in a mean increase of 3 nanomole / litre for every 100 international units of vitamin D, after adjusting for baseline 25 (OH)D concentration and country latitude. Controlling for country latitude was the marker used for sun exposure. The prevalence of vitamin D deficiency was lower with a risk ratio of 0.53, and cognitive function improved by a mean difference of 1.22 intelligence quotient points in healthy children aged between one and eighteen. Brandão-Lima PN et al reported an increase in the serum concentrations of 25(OH)D with the consumption of dairy fortified foods, in children aged between two and eleven, who were able to achieve or maintain vitamin D sufficiency status [36].

Although some 60% of our respondents said that they would prefer to have foods fortified with vitamin D instead of taking separate supplements, many respondents provided informative contextualised reasons on why they did not like the idea of eating processed or fortified foods (see Box 2). Reasons for preferring supplementation included worrying about not getting enough vitamin D from fortified food and that having fortified foods would not be as good as taking supplements, it would be easier to know how much you have taken if you 'just take a tablet' and 'far more convenient', and that it would be 'easier' to take a supplement with 'normal' non-fortified food, especially if fortified foods were those not usually part of 'your normal diet'. A particular issue that was flagged up was being unable to have fortified foods, as in the case of a diabetic having to control sugar intake or being a vegetarian or someone following particular diet plans, which would limit food options. Price of fortified foods was a concern, with respondent's saying that they did not 'want to buy products I wouldn't buy regularly' or 'it would limit my choosing' option and may affect price'.

In their global consensus publication, Munns CF et al make recommendations for the prevention and management of nutritional rickets; these include safe sun exposure, vitamin D supplementation, combined with the strategic fortification of normal habitual foods accompanied by adequate dietary calcium intake [5]. The fortification of milk with vitamin D in the 1930s effectively eradicated rickets worldwide [43–45]. However, research has also shown [46, 47], that fortification alone is not enough, and that additional vitamin D supplementation is needed to achieve sufficiency status in at-risk populations.

*Sun exposure*. With reference to sun exposure, our analysis found that skin coverage was significantly associated both with ethnicity (p-value 0.01) and skin colour (p-value 0.03) and vitamin D intake (p-value <0.01) (Table 3). Therefore, targeting at-risk groups, whether based on ethnicity or skin colour, maybe just as effective. Hakim OA et al [16] undertook a UK-based study, finding that despite consistently lower 25(OH)D levels in South Asian women, they were shown to synthesise vitamin D as efficiently as Caucasians when exposed to the same dose of ultra-violet radiation. The baseline level of vitamin D rather than ethnicity and skin tone influenced the amount of vitamin D synthesised [16]. An evidence technological assessment [48] included eight randomized trials of ultraviolet (UV)-B radiation (both artificial and solar exposure) and reported a positive effect on serum 25(OH)D concentrations; however, limitations included the overall fewer number of participants and heterogeneity when considering the exact UV-B dose and 25(OH)D assay used. Cranney A et al could not determine how the 25(OH)D levels varied by ethnicity, sunscreen use or latitude [48]. Using New Zealand for their population case study, Callister's research group [49] tried to evaluate whether ethnicity could be used as a determinant for sun exposure health promotion messaging. Ethnicity is a cultural construct, and with ethnic intermarriage, there is a possible weakening of the relationship between ethnicity and skin colour. They recommended that skin colour and other variables such as the season and time of day were criteria that should inform the discussion of risks of sun exposure rather than simply targeting ethnic groups and suggested that direct measures of skin type may be beneficial when assessing the risks and benefits of sun exposure [49]. There is a need for more work with ethnic minority groups on what level of sun exposure may be needed depending on the individual's skin colour and how this could be culturally facilitated. Simplistic targeting of public health messaging on sun exposure based on ethnicity may not be the way forward.

Vitamin D is not the solution for all ills, but deficiency is widespread and the evidence for primary and secondary prevention is good. From a public health perspective, this is a particularly relevant message for at-risk black and minority ethnic groups [5, 6]. A worldwide public health intervention that includes vitamin D supplementation in certain risk groups, with systematic vitamin D food fortification to avoid severe vitamin D deficiency in the general

population, has been proposed as an appropriate and acceptable public health strategy. With its rare side effects and relatively wide safety margin [5, 50], vitamin D supplementation can be an effective, inexpensive, and safe adjuvant therapy for those with confirmed deficiency. In addition to advice on supplementation aimed at at-risk groups, population-wide based public health advice needs to incorporate information relevant to intake of fortified foods and sun exposure. Facilitators would include clear labelling of fortified foods and providing information on appropriate supplement doses as well as safe sun exposure. The information should be available from health professionals and on the NHS website, which many of our respondents indicated were their go-to sources for information. Around 15% of our respondents use the internet or other media for information, and these forums can be useful for the community with poor literacy [24].

## Strengths and limitations

To our knowledge, this is the first study to explore knowledge, attitudes and practices around Vitamin D in the UK population following the national lockdowns during the COVID-19 pandemic. It included a simple study framework, using iteratively developed online survey questions facilitated via the Qualtrics platform, which allowed us to obtain data from a range of respondents, in a time efficient way. We had a large sample size of respondents with good representation of different ethnicities. A limitation that needs to be considered is the higher healthy, educated respondent profile, which could be due to greater social media literacy and mitigated by the fact that their responses will enable us to ensure care for a population with ill health. Findings from this exploratory study could inform the development of a larger and more complex study with support from local councils, primary care and voluntary organisations to reach the wider UK population. Including the survey questions translated in other languages would also help widen access.

## Conclusion

In summary, we were successful in eliciting views on knowledge, attitudes, and practices around vitamin D from a diverse community population, including a large proportion of black and minority ethnic individuals. This online survey provides evidence that public health messaging about the health effects of Vitamin D is being acknowledged. Around four in ten individuals in our sample were taking some form of Vitamin D supplements. This however leaves a majority not being aware of or not following national guidance. The survey also identified a number of facilitators and barriers for community vitamin D supplementation and fortification which can be used to improve future public health and individual healthcare advice. With regards to sun exposure, both skin colour and ethnicity criteria can be used to inform safe exposure. The community wanted information and guidance to help manage individual vitamin D status, especially for high-risk groups, and on balancing supplementation, food fortification and sun exposure. We have carried out further work using focus group and one to one interview settings, to help build on our knowledge around barriers and facilitators to vitamin D usage, which will be reported in a second paper.

## Supporting information

**S1 File. Online e-survey questionnaire.**
(DOCX)

**S2 File. Summary document tables—Facilitators and Barriers with Community supplementation of Vitamin D (FABCOM-D).**
(DOCX)

**S3 File. Table 4 univariable and multivariable association of demographic characteristics with Vit-D intake.**
(DOCX)

## Acknowledgments

Aakriti Shah, Sundar Siddarth, Ioanna Zimianiti, medical students at Imperial College, London for assistance with early background literature searches to inform design of questions for the online survey, piloting the survey for clarity and ease of understanding and working with the team for experiential learning and formative research skills to include the ethics approval process. Eva F Riboli-Sasco, Research Fellow, Self-Care Academic Research Unit, Imperial College London for internal peer review support.

## Author Contributions

**Conceptualization:** Nuttan Kantilal Tanna, Iman Webber, Austen El-Costa, Mitch Blair.

**Data curation:** Nuttan Kantilal Tanna, Manisha Karki, Iman Webber, Aos Alaa.

**Formal analysis:** Nuttan Kantilal Tanna, Manisha Karki, Mitch Blair.

**Investigation:** Nuttan Kantilal Tanna.

**Methodology:** Nuttan Kantilal Tanna, Austen El-Costa, Mitch Blair.

**Resources:** Austen El-Costa.

**Software:** Iman Webber.

**Supervision:** Mitch Blair.

**Validation:** Manisha Karki.

**Writing – original draft:** Nuttan Kantilal Tanna.

**Writing – review & editing:** Nuttan Kantilal Tanna, Manisha Karki, Austen El-Costa, Mitch Blair.

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
