## [Decision Letter · Decision Letter 0]

26 Apr 2023

PONE-D-23-00391Knowledge, attitudes, and practices associated with Vitamin D supplementation: A cross-sectional online community survey of adults in the UKPLOS ONE

Dear Dr. Nuttan Kantilal Tanna,

Thank you for submitting your manuscript to PLOS ONE. After careful consideration, we feel that it has merit but does not fully meet PLOS ONE’s publication criteria as it currently stands. Therefore, we invite you to submit a revised version of the manuscript that addresses the points raised during the review process.

We look forward to receiving your revised manuscript.

Kind regards,

Sidrah Nausheen, FCPS

Academic Editor

PLOS ONE

“This research received no specific grant from any funding agency in the public, commercial or not-for-profit sectors. AEO & IW are in part supported by the National Institute for Health and Care Research (NIHR) Applied Research Collaboration (ARC) North-West London. The views expressed are those of the authors and not necessarily those of the NHS or the NIHR or the Department of Health and Social Care”

Reviewers' comments:

Reviewer's Responses to Questions

**Comments to the Author**

1. Is the manuscript technically sound, and do the data support the conclusions?

Reviewer #1: Yes

Reviewer #2: Yes

2. Has the statistical analysis been performed appropriately and rigorously? 

Reviewer #1: Yes

Reviewer #2: Yes

3. Have the authors made all data underlying the findings in their manuscript fully available?

Reviewer #1: Yes

Reviewer #2: Yes

4. Is the manuscript presented in an intelligible fashion and written in standard English?

Reviewer #1: Yes

Reviewer #2: Yes

5. Review Comments to the Author

Reviewer #1: Dear Author,

This is an interesting study considering the need for vitamin D supplementation in adult population. Overall, it is well written and structured study with intelligible presentation and appropriate methodology.

Reviewer #2: Table 2 could be simplified for better understanding.

comprehensive KAP regarding vit D in community.

Good representation of minority groups at particular higher risk.

provides baseline information to policy makers.

6. PLOS authors have the option to publish the peer review history of their article (what does this mean?). If published, this will include your full peer review and any attached files.

Reviewer #1: No

Reviewer #2: No

---

## [Author Response · Author response to Decision Letter 0]

12 Jul 2023

Paper: Knowledge, attitudes, and practices associated with Vitamin D supplementation: A cross-sectional online community survey of adults in the UK

Dear Professor Sidrah Nausheen, Academic Editor, PLOS ONE

Thank you both to you and the peer reviewers for reviewing our paper ‘Knowledge, attitudes, and practices associated with Vitamin D supplementation: A cross-sectional online community survey of adults in the UK’, and the helpful comments. We appreciated the positive peer reviewer comments. 

We have addressed all the modifications requested, bar one, and look forward to the paper being published in PLOS ONE with a little assistance from the editorial team. We would appreciate help with formatting of Tables 2 and 3 in the manuscript, with proof for approval, before publication. 

With thanks

Nuttan Tanna

…………………………………………………………………………………………………………………………………………….. 

PLOS ONE Journal Reviewer #1: Dear Author,

This is an interesting study considering the need for vitamin D supplementation in adult population. Overall, it is well written and structured study with intelligible presentation and appropriate methodology.

Reviewer #2: Table 2 could be simplified for better understanding.

comprehensive KAP regarding vit D in community.

Good representation of minority groups at particular higher risk.

provides baseline information to policy makers.

Dear Peer Reviewers,

Thank you for your valuable time to peer review and comment on our paper. We are happy to note that you found the paper interesting and comprehensive, reporting results from a well-structured study design. This paper reports findings from the first phase of the FABCOM-D study. This was an online electronic survey available via the Qualtrics platform on a smartphone or computer. We were pleased that we managed to get good representation of ethnic minority respondents, with findings thereby applicable to the diverse UK wide community. We are currently analysing the dataset from the second qualitative phase which includes focus group discussions sessions and one-to-one interviews, with the view that this will generate additional information to help build on the first phase findings, and further inform our knowledge on facilitators and barriers associated with vitamin D supplementation. 

We have simplified Table 2 and present the data within Tables 2 and 3 in the revised manuscript and hope that this helps with a better understanding of the statistical analysis and results. Table 4, available as a supplementary file, provides the dataset from the logistic regression analysis. 

Reviewer’s comments

1. The study objective is about barriers and drivers of vitamin D supplementation in defined population. The result is more focused towards association of risk factor to vitamin D usage (table 2). Either this should be added in the objectives or deleted from the results. 

Thank you for this important observation. To clarify, our defined population was the community, so those that would generally be able to self-care. The FABCOM-D study includes a study design with 2 phases. This paper reports findings from the first quantitative phase. A background literature search helped with identification of risk factors that needed to be considered when exploring for facilitators or barriers within the community setting. We have stated that the main outcome measures for the first phase were awareness of vitamin D sources, health benefits and testing and attitudes to supplementation, sun exposure and fortification. 

We have also simplified Table 2 and present the dataset within Tables 2 and 3 in the revised manuscript. 

2. It will be more impactful if a table is added showing which barriers and which drivers were identified for vitamin D usage in the study population. 

Thank you for this helpful comment. We are undertaking further work, with analysis of the dataset from phase 2. This second qualitative phase generated rich data from focus group discussions and individual interviews. The findings should add to and build on knowledge on facilitators and barriers around Vitamin D supplementation. We will report these findings in our next paper, in both narrative and tabular formats.

---

## [Editor Report · Decision Letter 1]

21 Jul 2023

Knowledge, attitudes, and practices associated with Vitamin D supplementation: A cross-sectional online community survey of adults in the UK

PONE-D-23-00391R1

Dear Dr.Nuttan Kantilal Tanna,

We’re pleased to inform you that your manuscript has been judged scientifically suitable for publication and will be formally accepted for publication once it meets all outstanding technical requirements.

Kind regards,

Sidrah Nausheen, FCPS

Academic Editor

PLOS ONE
---

## [Editor Report · Acceptance letter]

28 Jul 2023

PONE-D-23-00391R1 

Knowledge, attitudes, and practices associated with Vitamin D supplementation: A cross-sectional online community survey of adults in the UK 

Dear Dr. Tanna:

I'm pleased to inform you that your manuscript has been deemed suitable for publication in PLOS ONE. Congratulations! Your manuscript is now with our production department. 

Kind regards, 

on behalf of

Dr. Sidrah Nausheen 

Academic Editor

PLOS ONE